# Evidence of Isomerization in the Michael-Type Thiol-Maleimide Addition: Click Reaction between L-Cysteine and 6-Maleimidehexanoic Acid

**DOI:** 10.3390/molecules27165064

**Published:** 2022-08-09

**Authors:** Víctor Alfonso Niño-Ramírez, Diego Sebastián Insuasty-Cepeda, Zuly Jenny Rivera-Monroy, Mauricio Maldonado

**Affiliations:** Chemistry Department, Universidad Nacional de Colombia, Bogotá, Carrera 45 No 26-85, Building 451, Office 409, Bogotá 11321, Colombia

**Keywords:** click reaction, Michael addition, thiol-maleimide addition, isomerization

## Abstract

The reaction between L-cysteine (Cys) and 6-maleimidohexanoic acid (Mhx) in an aqueous medium at different levels of pH was analyzed via RP-HPLC, finding the presence of two reaction products throughout the evaluated pH range. By means of solid-phase extraction (SPE), it was possible to separate the products and obtain isolated profiles enriched up to 80%. The products were analyzed individually through mass spectrometry, DAD-HPLC, NMR ^1^H, ^13^C, and two-dimensional evidence of isomerization between the hydrogen atoms of the α-amino and the thiol group present in the cysteine. Thus, it was concluded that the products obtained corresponded to a mixture of the isomer Cys-S-Mhx, where the adduct is formed by a thioether bond, and the isomer Cys-NH-Mhx, in which the union is driven by the amino group. We consider that the phenomenon of isomerization is an important finding, since it has not previously been reported for this reaction.

## 1. Introduction

The Michael-type thiol-maleimide addition reaction is one of the most versatile in biochemistry, organic synthesis, and bioconjugation, thanks to its rapid kinetics, quantitative conversion, selectivity, high yields, and lack of by-products. This reaction is classified within the thiol-ene type reactions, where the maleimide group in particular exhibits great reactivity compared to other ene groups, such as vinyl sulfones, acrylates, acrylamides, methacrylates, etc., which have poor reactivity given the structure and groups adjacent to the active ene, and lower reaction kinetics compared to maleimide [1,2,3]. This addition reaction is very versatile; depending on the physicochemical and structural characteristics of the starting molecules functionalized with free thiol or maleimide groups, it can be used with aqueous or organic solvents, both protic and aprotic, allowing a very wide range of options for solubilizing the different functionalized precursors. Additionally, since there are two reaction mechanisms catalyzed by bases or by nucleophiles, it is possible to explore the thiol-maleimide addition reaction in the absence of a catalyst, or by initiators, although it is typically used with tertiary amines [1,4,5].

The use of this click reaction can be found in various examples, including: (i) Incorporation of peptide motifs that allow the construction of antibody–drug conjugates (ADCs) [6,7]; (ii) Coupling of Pt(IV) coordination compounds with maleimide and linear peptide molecules [8,9]; (iii) Design and synthesis of nanoparticles such as molecular cages or nanoparticle drug delivery systems [10,11]; (iv) Construction of dendrimeric molecules [12]; (v) Obtaining oligonucleotides-peptide [13]; (vi) Functionalization of a monolithic organic support, poly(GMA*-co-*EDMA), with a maleimide hexanoic (Mhx) group that allows the incorporation of a peptide containing a residue of cysteine at the N-terminal, a reaction that was found to be selective under mild conditions [14].

Despite the benefits of this reaction, it has been reported that the thiosuccinimide resulting from the thiol-maleimide reaction may be hydrolytically unstable through a retro-Michael reaction (Figure 1), [2,6,15]. In particular, although the presence of retro-Michael reactions can be seen as a disadvantage, it makes it possible to vary the stability of the succinimide thioether adduct. Several studies have shown how the reversible reaction and N-substituting groups can have an effect on stability, and therefore allow proper tuning and release according to environmental conditions, especially in the case of ADCs [16]. In addition, other researchers have pointed out that the formation of a single thioether succinimide product may occur, while still others suggest the possible presence of two diastereoisomeres. Specifically, the diastereoisomers correspond to the addition of two adducts with stereocenters R and S into the thiol group on the α, β-unsaturated bond. However, if the ring of the resulting thiosuccinimidil is opened by hydrolysis, mixtures of succinamic acid are generated, although the advantage is that a retro Michael thiol-maleimide reaction is no longer possible, obtaining a completely stable adduct [3,17]. Additionally, as noted above, a third subsequent reaction can be caused by the addition of thiol-maleimide, forming a new adduct with a six-member cycle called thiazine [9]. Figure 1 summarizes the reported behavior of this addition reaction.

Considering the great applicability and versatility of this reaction, and its relevance to the field of biochemistry and particularly the construction of conjugate systems, we used this reaction to join the 6-maleimidehexanoic acid to a peptide which contained a N-terminal cysteine. This reaction was monitored via RP-HPLC, and it was observed that the chromatographic profile presented two signals corresponded to addition reaction products [14]. In this context, the thiol-maleimide addition was studied in the present investigation, using as a model the reaction between L-Cysteine (Cys) and 6-maleimide hexanoic acid (Mhx) under different experimental conditions. These precursors were selected due to our interest in using the thiol-maleimido Michael addition reaction for the construction of different conjugates between peptide sequences containing the L-cysteine or 6-maleimidohexanoic motif, and non-peptidyl molecules.

## 2. Results and Discussion

As mentioned above, the high reactivity of maleimides is caused by their behavior as an electrophile, due to the α,β-unsaturation and the presence of two carbonyl groups, which allows them to react quickly with nucleophiles such as thiolate groups, obtaining succinimide adducts. Furthermore, the addition of Michael thiol-maleimide is highly specific to thiols at pH 6.5–7.5, and is 1000 times faster than the slow addition that would occur in the presence of ε-amino groups of Lys at pH 7.0, due to the nucleophilia of the thiolate group and the pKa of the thiol group [7]. Therefore, to explore the reactivity, selectivity, and conditions of the click reaction between cysteine and maleimidehexanoic acid, the process was initially begun through Michael addition in water at room temperature, in a manner similar to that described in the literature [18,19,20]. After 2 h, the reaction mixture was analyzed by means of LC-MS (ESI) and RP-HPLC. As shown in Figure 2, the reaction’s chromatographic profile showed four signals, corresponding to cysteine (Cys, t_R_ = 0.92 min), 6-maleimidohexanoic acid (Mhx, t_R_ = 7.56 min), and two products at t_R_ = 4.6 and 4.7 min, which showed similar signals in their MS spectra, at *m*/*z* 333.1118 and 333.1117, respectively. For the species [M + H]^+^ the calculated exact mass corresponded to 333.1120, and therefore, in both cases the error was less than 1 ppm (Figure 2).

The chromatographic method for the analysis of the reaction by RP-HPLC (DAD) was improved, and it was possible to separate the two reaction products with a resolution of two. The RP-HPLC-DAD analysis, in the range of 190 to 270 nm, allowed determination of the purity of each product. The two peaks had a similar UV spectrum. Product 1 had a peak purity of 99.7%, and Product 2 had a peak purity of 99.8% (Figure 3A, in water). Then the reaction was carried out by varying the reaction conditions, such as solvent type and pH (Figure 3), and the reaction mixtures were analyzed using the optimized RP-HPLC method.

As can be seen, the reaction was favored in protic solvents. When methanol or water were used, the reaction was quantitative and occurred in time spans of less than 5 min. The Mhx peak disappeared completely, and the two reaction products previously described were formed (peaks 1 and 2). When the polarity of the solvent began to decrease, as in the case of ethanol, the appearance of reaction products was observed at a rather low proportion, and the Mhx peak was observed (peak 3). It is possible that solvation phenomena or the effect of the polarity of the solvent reduced the yield of the reaction. Finally, in the case of acetonitrile (ACN), the reaction did not take place; no reaction products were seen and only the Mhx peak appeared in the chromatogram. The reaction in the aqueous phase was evaluated by varying the pH from 2.5 to 6.8, and in all cases, there was the rapid appearance of the two previously observed reaction products. This result suggests that the reaction took place over a wide pH spectrum.

The unexpected formation of two products by means of the methodology adapted from the literature contrasts with results obtained previously [3,16,21], in which it was reported that thiomethylation was the only reaction to occur. The formation of two products with the same mass, as evidenced in the chromatogram, is an interesting result. We considered two possible explanations, viz., the formation of a diasteroisomer mixture or of a tautomer mixture. To establish which of the two reaction routes was used, we proceeded to separate the two products via the RP-SPE technique [22] (see the experimental section). The isolated products exhibited a great tendency to be hygroscopic; for this reason, their characterization was only carried out in solution.

The mixture and the purified products were analyzed by means of NMR spectroscopy. The analysis of the ^1^H-NMR spectrum of the unpurified mixture showed two products corresponding to the isomeric mixture that presented well-resolved signals in the aliphatic region, which allowed easy assignment of the two isomers and even their molar ratios in the crude product (50/50) (Figure 4). After separating the isomers, the NMR analysis of each of the products obtained was carried out.

The ^1^H-NMR spectra, in Methanol-d_6_, of the more polar compound (peak 1 in Figure 3) showed three signals, at 3.14, 3.70, and 4.23 ppm, corresponding to the protons in the maleimide ring. The signals corresponding to the cysteine residue were seen at 4.02 for the alpha proton and at 3.25 and 2.55 for the diasterotopic protons. Additionally, the ^1^H-NMR spectrum displayed characteristic signals of hexanoic acid proton residue at 1.34, 1.60, 2.29, and 3.51 ppm. The ^13^C-NMR spectrum showed thirteen signals, which were unambiguously assigned through 1D- and 2D-NMR experiments, including the HMQC and HMBC spectra. The signals for carbon atoms linked to hydrogen atoms were assigned based on the observed correlations in the HMQC spectrum. The signals that did not show any correlation in the HMQC spectrum were those that were not linked to protons. Thus, all the patterns were consistent with the expected Michael thiol-maleimide addition product, Compound **1** (Cys-S-Mhx, Figure 4). On the other hand, the spectrum of the second product, Compound **2**, displayed essentially the same signals for the cysteine residue and for the hexanoic acid residue; however, the signals for the protons of the maleimide ring showed certain differences, particularly in their chemical shifts. In this way, three different proton moieties for the maleimide ring, with two signals at 3.29 and 3.48 ppm assigned to the methylen group in the maleimide ring, and one signal at 4.40 ppm corresponding to the methyne proton in the maleimide ring, indicated that the ring was attached to the nitrogen of the cysteine residue. Like Compound 1, the ^13^C-NMR spectrum of the second product showed thirteen signals, which were unambiguously assigned through ^1^D- and ^2^D-NMR experiments, including the HMQC and HMBC spectra. In this way, the signal at 52.89 ppm allowed confirmation of the connectivity of the cysteine with the maleimide ring by means of nitrogen; additionally, this signal was not observed in the ^13^C-NMR spectrum of Compound **1**. The complete signal assignment and the observed correlations for Compound **2** are shown in Table 1.

Based on this information from NMR and from the mass spectra, the products formed corresponded to a mixture of tautomers. The formation of the two isomers can be explained as follows. Initially, the reaction of the Michael addition, which took place by means of nucleophilic attack of the thiol group of the cysteine on the maleimide ring, generated Compound **1**. Then, the presence of an amino group in the cysteine residue favored an intramolecular nucleophilic substitution of the thioether by the amino group, forming a cyclic intermediary, which finally generated Compound **2**, Cys-NH-Mhx, as shown in Figure 5, as described in previous papers [3,17].

To corroborate the results obtained by NMR and MS that pointed toward the formation of tautomers, a final study was carried out in which the amino acid Fmoc-Cys-OH was used. If the reaction products correspond to tautomers, when the Fmoc-Cys-OH reacts with the Mhx, the amino group is blocked by the Fmoc group, so tautomerism is not possible and a single reaction product should appear. This reaction was carried out in an aqueous medium and was monitored by means of RP-HPLC. As can be seen (Figure 6, peak 3), at 6.75 min a single product was observable. This result strengthens the hypothesis that the products corresponded to tautomers.

Finally, to establish if this behavior was evidenced with other substrates, we decided to observe the reaction between L-cysteine and *N*-(2-aminoethyl)maleimide under the same experimental conditions (H_2_O/MeOH 80:20). As can be seen in Figure 7, two peaks with very similar retention times were observed in the chromatogram, allowing the conclusion that the reaction proceeded to form the same types of products.

## 3. Materials and Methods

### 3.1. Reagents and Materials

L-Cysteine, 6-maleimidohexanoic acid, trifluoroacetic acid (TFA), acetonitrile (ACN), methanol (MeOH), and ethanol (EtOH) were purchased from Sigma-Aldrich (St. Louis, MO, USA). Methanol-*d_6_* was obtained from Merck (Darmstadt, Germany).

### 3.2. General Procedure for the Reaction of Cysteine and 6-Maleimidohexanoic Acid

14.6 mg of L-cysteine and 28.2 mg of 6-maleimidohexanoic acid were mixed in water-methanol 95:5 at pH 6.8 and stirred for 3 min. The mixture was analyzed via RP-HPLC, the two products obtained were purified and enriched via SPE, and were analyzed individually via RP-HPLC, ESI-MS, NMR ^1^H, ^13^C, and two-dimensional NMR. ^1^H-NMR and ^13^C-NMR spectra were recorded at 400 MHz on a Bruker Avance 400 instrument. Chemical shifts were reported in ppm, using the solvent residual signal. Molar mass was determined on a Thermo Scientific HPLC-MS, RP–HPLC analyses were performed using an Agilent 1200 Liquid Chromatograph (Agilent, Omaha, NE, USA), and for the DAD study a Thermo Scientific Ultimate 3000 HPLC was used. The product was characterized via ^1^H NMR, ^13^C NMR, MS, and HPLC. The following was obtained:

6-(3-((2-amino-2-carboxyethyl)thio)-2,5-dioxopyrrolidin-1-yl)hexanoic acid (1) ^1^H-NMR, Methanol*-d_4_*, δ (ppm): 1.26 (d, 12H, *J* = 8Hz, CH3), 4.42 (q, 4H, *J* = 8Hz, CH), 6.11 (s, 4H, H *orto to* OH), 6.73 (s, 4H, H *meta* to OH), 8.53 (s, 8H, OH); ^13^C-NMR, Methanol*-d_4_*, δ (ppm): 20.3; 29.1; 104.0; 124.8; 126.4; 152.7. ESI–TOF/MS analysis showed a signal at *m/z* = 333.1118 corresponding to [M + H]^+^, calculated *m*/*z* 333.1120 

6-(3-((1-carboxy-2-mercaptoethyl)amino)-2,5-dioxopyrrolidin-1-yl)hexanoic acid (2) ^1^H-NMR, DMSO*-d_6_*, δ (ppm): 0.85 (t, 12H, *J* = 7Hz, CH_3_), 1.19 (m, 8H, CH_2_), 1.28 (m, 16H, CH_2_), 2.03 (d, 8H, CH_2_) 4.24 (t, 4H, *J* = 6.8 Hz, CH), 6.17 (s, 4H, H *orto* to OH), 7.16 (s, 4H, H *meta* to OH), 8.90 (s, 8H, OH). ^13^C-NMR, DMSO*-d_6_* δ (ppm): 14.1; 22.4; 27.6; 31.6; 33.1; 34.1; 102.5; 123.2; 124.9; 151.8. ESI–TOF/MS analysis showed a signal at *m/z* = 333.1117 corresponding to [M + H]^+^, calculated *m*/*z* 333.1120

### 3.3. Reverse-Phase High-Performance Liquid Chromatography (RP-HPLC) Analysis

For the reaction products between 6-maleimidohexanoic acid and L-cysteine RP-HPLC analysis, 10 µL solution of each evaluated reaction condition was injected. A linear gradient from 5% to 50% of solvent B (ACN-TFA 0.05%) in solvent A (H_2_O-TFA 0.05%) on a monolithic Chromolith^®^ C-18 column (Merck KGaA, Darmstadt, Germany, 50 × 100 mm) for 8 min at a wavelength of 210 nm was used. For the DAD study, a Thermo Scientific Ultimate 3000 DAD-HPLC was used, with 10 µL of the Cys–Maleimido reaction mixture taken in H_2_O and analyzed by means of the linear gradient previously described, using a Thermo Scientific Eclipse C-18 packed column (100 mm ×4.5 µm). The reaction peaks, at a length of 190–270 nm, were analyzed by means of DAD.

### 3.4. RP-SPE Purification

To enrich the reaction products obtained in the reaction, the method previously described by Insuasty et al. was used [22]. Briefly, an RP-SPE (Supelclean^TM^) column was activated with 30 mL of methanol, 30 mL of ACN-TFA (0.05%), and 30 mL of H_2_O-TFA (0.05%). Subsequently, 1 mL of solution containing the mixture obtained in the previous section was added, and the elution of the products was performed by increasing the quantity of solvent B in the eluent by 12 mL fractions. Analysis was completed by means of RP-HPLC, and the fractions in which the reaction products were more enriched were frozen and later lyophilized for storage.

### 3.5. LC-MS Methodologies

The Cys–Mal mixture reaction samples were prepared according to the previous general procedure. The mixture reaction was centrifuged at 15,000 rpm for 3 min at room temperature, the supernatant was diluted 1000 times, and 2 µL was analyzed in a Bruker Impact II LC Q-TOF MS equipped with electrospray ionization (ESI) in positive mode. The chromatographic conditions were maintained with an Intensity Solo C18 column (2.1 × 100 mm, 1.8 μm) (Bruker Daltonik, Billerica, MA, USA), at a temperature of 40 °C and a flow rate of 0.250 mL min^−1^. Mobile phase water (A) and acetonitrile (B) were used, each containing 0.1% formic acid. Gradient elution was 5/5/95/95/5/5%B at 0/1/11/13/13.1/15 min. ESI source conditions: end plate offset 500 V, capillary 4500 V, nebulizer 1.8 bar, dry gas nitrogen 8.0 L/min, dry temp 220 °C. Scan mode AutoMS/MS with spectral range 20–1000 *m*/*z*, spectra rate 2 Hz, collision energy 5.0 eV (See Appendix A).

## 4. Conclusions

The reaction between Cys and Mhx is a rapid and quantitative reaction that is favored in protic solvents and occurs over a wide pH range (2.5 to 8), under mild conditions. Our research unexpectedly found that this reaction generated two reaction products, and by means of NMR, MS, and HPLC, it was possible to verify that these corresponded to the tautomers Cys-S-Mhx, where a rearrangement of the S and N atoms occurred. This is an interesting finding, since the Michael reaction between the thiol group and the maleimide group is a well-known and widely-used reaction with various biochemical applications, but up to now the formation of two reaction products corresponding to tautomers had never been reported.

## Figures and Tables

**Figure 1 molecules-27-05064-f001:**
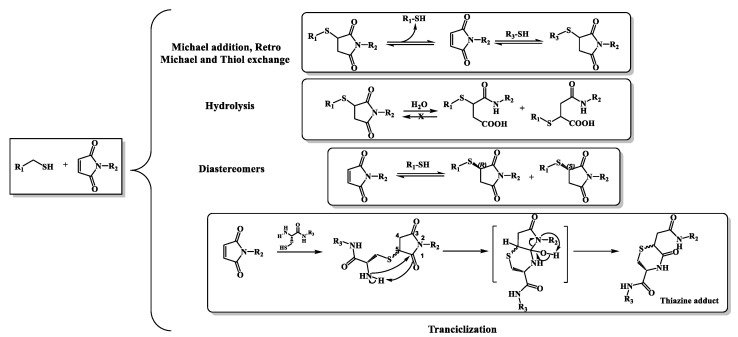
Michael thiol-maleimide addition reaction, rearrangements and products formed.

**Figure 2 molecules-27-05064-f002:**
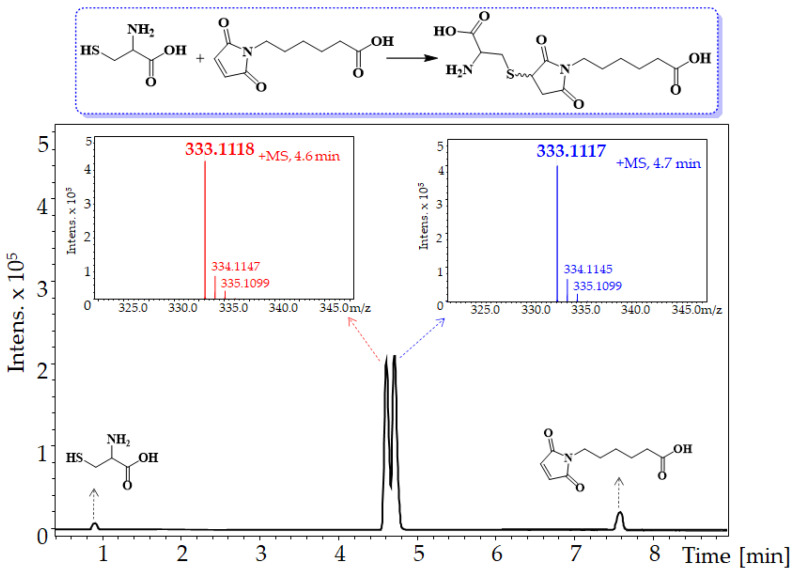
Analysis of the reaction between cysteine (Cys) and 6-maleimidohexanoic acid (Mhx) in water, by means of LC-MS (ESI). The TIC showed signals at t_R_**:** 4.6 and 4.7 min that corresponded to the reaction’s products.

**Figure 3 molecules-27-05064-f003:**
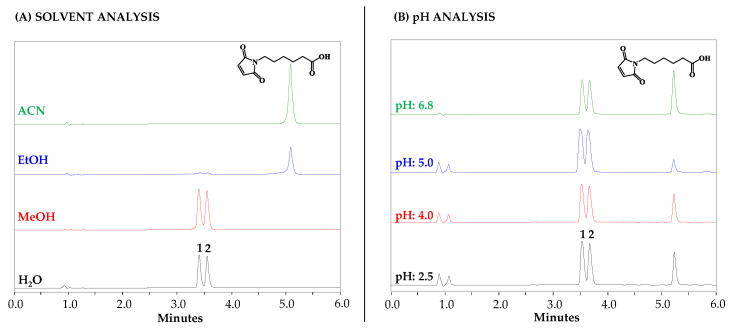
RP-HPLC monitoring of the reaction between Cys and maleimidehexanoic acid (Mhx). (**A**) Reaction in different solvents (H_2_O, MeOH, EtOH, and ACN). (**B**) Reaction in aqueous solution at different pH values: 2.5, 4.0, 5.0 and 6.8. Peaks 1 and 2 correspond to the reaction products.

**Figure 4 molecules-27-05064-f004:**
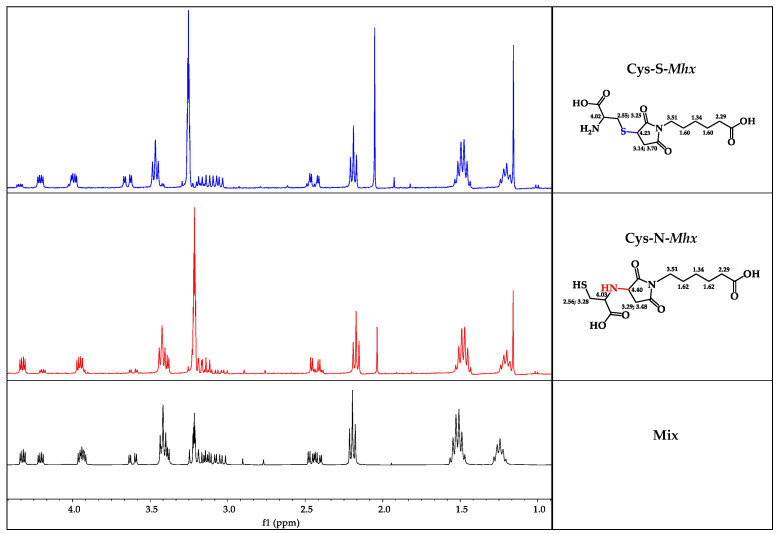
Comparison of NMR spectra in DMSO*-d_6_*, for mixtures of products and individual adducts.

**Figure 5 molecules-27-05064-f005:**
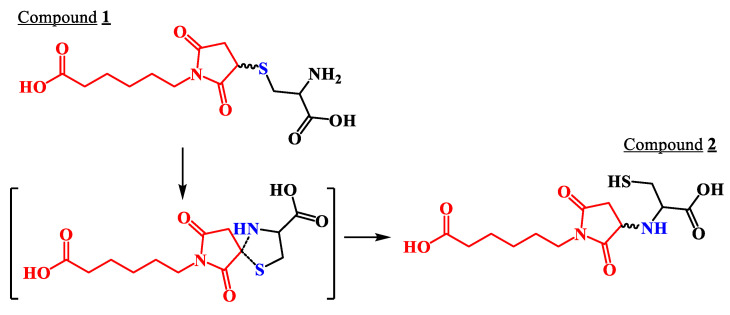
Possible formation of the cyclic intermediate between Compounds **1** and **2**.

**Figure 6 molecules-27-05064-f006:**
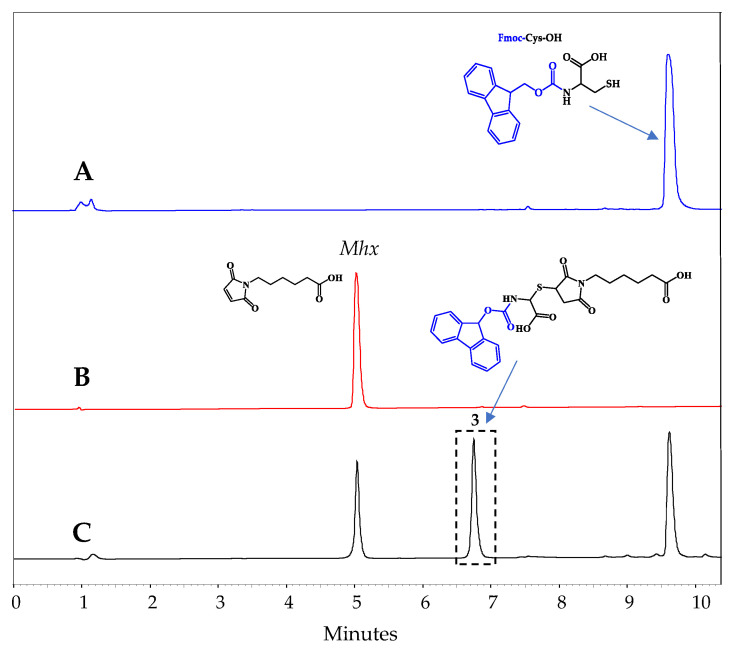
Fmoc-Cys-OH and Mhx reaction. Monitoring by RP-HPLC: panels (**A**,**B**) correspond to starting material, panel (**C**) shows the reaction mixture.

**Figure 7 molecules-27-05064-f007:**
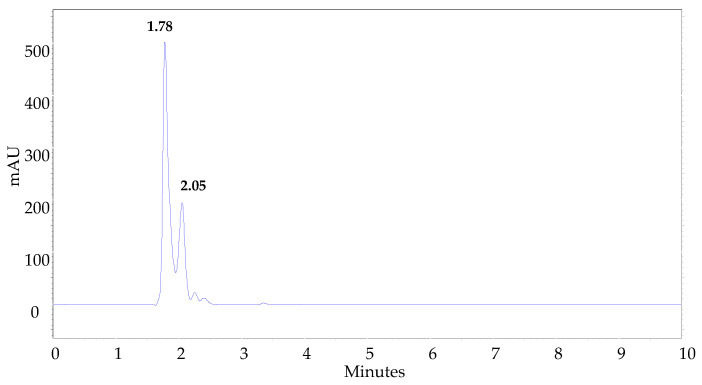
Chromatogram of the reaction mixture between L-cysteine and *N*-(2-aminoethyl)maleimide in H_2_O/MeOH 80:20.

**Table 1 molecules-27-05064-t001:** Correlations for Compound **2**.

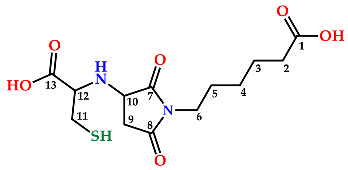			
Carbon	δ (ppm)	Correlation HMQC	Correlation HMBC
1	175.99	---	1.62, 2.29
2	33.22	2.29	--
3	26.73	1.62	--
4	25.80	1.34	--
5	26.73	1.62	--
6	38.46	3.51	--
7	178.41	---	3.51
8	174.95	---	2.56, 3.29, 3.51
9	32.99	3.29, 3.48	--
10	52.89	4.40	--
11	35.27	2.56, 3.28	--
12	41.09	4.03	2.56, 3.29
13	168.98	---	--

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
