# Peer review of "Evidence of Isomerization in the Michael-Type Thiol-Maleimide Addition: Click Reaction between L-Cysteine and 6-Maleimidehexanoic Acid"

_molecules, 2022, doi:10.3390/molecules27165064_

Round 1

Reviewer 1 Report

This revised manuscript submitted by Ramirez, Cepeda, Monroy, and Maldonado was greatly improved. Most of my concerns about the previous version have been addressed. However, I still think this paper lacks mechanistic study. Instead of using HPLC, the authors should consider using NMR to monitor the reaction, giving more information about the intermediate and the time scale of the reaction. It's great that the authors have shown that they observe this isomerization from other substrates, but I think the authors should give more characterization data in that section. Also, I think that can explore more simpler substrates.

Author Response

12th July 2022.

Dear Editor of MOLECULES,

We are sending the corrected manuscript of the article, “Evidence of isomerization in the Michael-type thiol-maleimide addition: Click reaction between L-cysteine and 6-maleimidehexanoic acid.” by Víctor Alfonso Niño Ramírez, Diego Sebastián Insuasty Cepeda, Zuly Jenny Rivera Monroy and Mauricio Maldonado, it was modified according with the Reviewers comments.

Following you will find, in blue, the modifications made in the manuscript as well as the answers to each Reviewer comments.

Yours sincerely,

Professor

Zuly Rivera Monroy

Chemistry Department, Sciences Faculty

Universidad Nacional de Colombia

Carrera 45 No 26-85, Bogotá-Colombia

Phone: +57 (1) 3165000 ext 14408/ +57 300 747 9639

Review report 1

This revised manuscript submitted by Ramirez, Cepeda, Monroy, and Maldonado was greatly improved. Most of my concerns about the previous version have been addressed. However, I still think this paper lacks mechanistic study. Instead of using HPLC, the authors should consider using NMR to monitor the reaction, giving more information about the intermediate and the time scale of the reaction. It's great that the authors have shown that they observe this isomerization from other substrates, but I think the authors should give more characterization data in that section. Also, I think that can explore more simpler substrates.

Answer. Thanks for the reviewer comment, we agree with it. However, we found that the reaction is very sensitive to pH, and controlling this variable in the conditions for obtaining the NMR spectrum is really complex.

Reviewer 2 Report

Given paper is devoted to the study of the reaction between cysteine and 6-maleimidehexanoic acid. This rather pedestrian reaction surprisingly ends up with the formation of two isomeric products. This kind of reactivity was never reported before despite wide application of Michael-type thiol-maleimide addition. The paper is well written and could be potentially considered for publication after some minor issues are addressed:

Figure 1. Bottom raw. I would suggest to flip products structures so the imide moiety will be to the right. Like it is in other structures.

Lines86-89: “the high reactivity of maleimides is due to their behavior as a highly reactive electrophile, thanks to the α, β-electrodeficient unsaturation and the presence of two carbonyl groups, which allows them to react quickly with nucleophiles such 88 as thiolate groups, allowing the obtaining of succinimide adducts” – I recommended to rephrase this sentence. It is a weird sequence of reason because of reason because of reason.

I highly doubt the polarity difference in MeOH and Et is so dramatic to change the reactivity that much. Did authors consider other possible effects such as solvation?

HRMS data for obtained compounds is missing.

Melting points of obtained compounds are missing.

Even though the article in genneral is fine, presented results are not impactful enough for the publication in Molecules. Based on the abovementioned I would recommend publishing this paper in other journal due to the rather poor research results.

Author Response

12th July 2022.

Dear Editor of MOLECULES,

We are sending the corrected manuscript of the article, “Evidence of isomerization in the Michael-type thiol-maleimide addition: Click reaction between L-cysteine and 6-maleimidehexanoic acid.” by Víctor Alfonso Niño Ramírez, Diego Sebastián Insuasty Cepeda, Zuly Jenny Rivera Monroy and Mauricio Maldonado, it was modified according with the Reviewers comments.

Following you will find, in blue, the modifications made in the manuscript as well as the answers to each Reviewer comments.

Yours sincerely,

Professor

Zuly Rivera Monroy

Chemistry Department, Sciences Faculty

Universidad Nacional de Colombia

Carrera 45 No 26-85, Bogotá-Colombia

Phone: +57 (1) 3165000 ext 14408/ +57 300 747 9639

Review report 2

Given paper is devoted to the study of the reaction between cysteine and 6-maleimidehexanoic acid. This rather pedestrian reaction surprisingly ends up with the formation of two isomeric products. This kind of reactivity was never reported before despite wide application of Michael-type thiol-maleimide addition. The paper is well written and could be potentially considered for publication after some minor issues are addressed:

Figure 1. Bottom raw. I would suggest to flip products structures so the imide moiety will be to the right. Like it is in other structures.

Answer. We agree with the reviewer. Figure 1 was corrected according with the reviewer comment

Lines86-89: “the high reactivity of maleimides is due to their behavior as a highly reactive electrophile, thanks to the αβ-electrodeficient unsaturation and the presence of two carbonyl groups, which allows them to react quickly with nucleophiles such 88 as thiolate groups, allowing the obtaining of succinimide adducts” – I recommended to rephrase this sentence. It is a weird sequence of reason because of reason because of reason.

Answer. We agree with the reviewer. It was corrected (lines 91-94)

I highly doubt the polarity difference in MeOH and Et is so dramatic to change the reactivity that much. Did authors consider other possible effects such as solvation?

Answer.  We agree with the reviewer. We include the following sentence to clarify “It is possible that the effect of the polarity of the solvent or solvation phenomena decrease the yield of the reaction.” (lines 125-126)

HRMS data for obtained compounds is missing.

Answer. We agree with the reviewer. We include the information in Figure 2 and also in lines 236 and 244

Melting points of obtained compounds are missing.

Answer. We agree with the reviewer. however, because these compounds are very hygroscopic, it was not possible to determine the melting points.

Even though the article in general is fine, presented results are not impactful enough for the publication in Molecules. Based on the abovementioned I would recommend publishing this paper in other journal due to the rather poor research results.

Answer. Thanks for the comment. However, we think that this paper is important for research groups that use this reaction as a synthetic tool for obtaining bioconjugates.

Reviewer 3 Report

1)    It is recommended to include a list of abbreviation at the start of the manuscript.

2)    Introduction: The focus objective of this research work should clearly stated. The justification for the selection for the combination of this 2 chemicals is not clear. What is the novelty of this research project?

3)    Figure 2 on Pg 3 is small & resolution needs to be improved

4)    There are 2 different Figure 6 in the manuscript [refer pg 5 & 6]. Subsequently, this leads to incorrect mention of the figures in the manuscript. Please check carefully.

5)    The so-called Figure 6 on Pg 7 is small & resolution needs to be improved.

6)    Line 198, Pg 7 & Line 203: ‘L-cisteina’ is not in English. T-(aminoethyl)maleimide should be capitalized in both lines.

7)     Section 4.2, Pg 7, line 226, 231, 232: CH2 and CH3 should be correctly formatted. 

8)     Section 4.2, Pg 7, line 226: 4,42 (c, 4H,  J = 8HZ, CH): what does ‘c’ refers to here?

9)    Conclusion need to be improved to highlight the objective of research work and show the significance/importance of the research work.

10)    References: Formatting for references ought to follow MDPI guideline. For this topic, there could be more than 29 relevant references relevant and closely related to this research topic.

Author Response

12th July 2022.

Dear Editor of MOLECULES,

We are sending the corrected manuscript of the article, “Evidence of isomerization in the Michael-type thiol-maleimide addition: Click reaction between L-cysteine and 6-maleimidehexanoic acid.” by Víctor Alfonso Niño Ramírez, Diego Sebastián Insuasty Cepeda, Zuly Jenny Rivera Monroy and Mauricio Maldonado, it was modified according with the Reviewers comments.

Following you will find, in blue, the modifications made in the manuscript as well as the answers to each Reviewer comments.

Yours sincerely,

Professor

Zuly Rivera Monroy

Chemistry Department, Sciences Faculty

Universidad Nacional de Colombia

Carrera 45 No 26-85, Bogotá-Colombia

Phone: +57 (1) 3165000 ext 14408/ +57 300 747 9639

Review report 3

1)    It is recommended to include a list of abbreviation at the start of the manuscript.

Answer. Thanks for the comment. We use other format, in which each abbreviation is indicated the first time it is used

2)    Introduction: The focus objective of this research work should clearly stated. The justification for the selection for the combination of this 2 chemicals is not clear. What is the novelty of this research project?

Answer. We agree with the reviewer. We expand the justification to clarify (lines 71-81): Considering the great applicability and versatility of this reaction relevant to the field of biochemistry and the construction of conjugate systems, we had use this reaction to join the 6-maleimidehexanoic acid to a peptide which contained a N-terminal cysteine, this reaction was monitored via RP-HPLC, and it was observed that the chromatographic profile presented two signals corresponding to addition reaction products [14]. In this context, in the present investigation the thiol-maleimide addition was studied, using as a model the reaction between L-Cysteine (Cys) and 6-maleimide hexanoic acid (Mhx) under different experimental conditions. These precursors were selected due to our interest in using the thiol-maleimido Michael addition reaction for the construction of different conjugates between peptide sequences containing the L-cysteine or 6-maleimidohexanoic motif and non-peptidyl molecules.

3) Figure 2 on Pg 3 is small & resolution needs to be improved

Answer. We agree with the reviewer. The figure 2 was corrected

4) There are 2 different Figure 6 in the manuscript [refer pg 5 & 6]. Subsequently, this leads to incorrect mention of the figures in the manuscript. Please check carefully.
Answer. We agree with the reviewer. It was corrected
5) The so-called Figure 6 on Pg 7 is small & resolution needs to be improved.
Answer. We agree with the reviewer. It was corrected
6) Line 198, Pg 7 & Line 203: ‘L-cisteina’ is not in English. T-(aminoethyl)maleimide should be capitalized in both lines.
Answer. We agree with the reviewer. It was corrected
7)Section 4.2, Pg 7, line 226, 231, 232: CH2 and CH3 should be correctly formatted. 
Answer. We agree with the reviewer. It was corrected (lines 238-239)
8)Section 4.2, Pg 7, line 226: 4,42 (c, 4H,  J = 8HZ, CH): what does ‘c’ refers to here?
Answer. We agree with the reviewer. It was corrected (line 233)
9) Conclusion need to be improved to highlight the objective of research work and show the significance/importance of the research work.

Answer. We agree with the reviewer. The objective of the work was indicate at introduction, and it is according with our conclusions

10) References: Formatting for references ought to follow MDPI guideline. For this topic, there could be more than 29 relevant references relevant and closely related to this research topic.

Answer. We agree with the reviewer. It was corrected

Round 2

Reviewer 2 Report

The authors considered all my criticism and made proper changes to the paper. I have no other criticism or questions left. However, my overall opinion stays – this research has very small result and is not suitable for Molecules. Otherwise it can be published as it is.

Author Response

Thanks for your revision and comments

This manuscript is a resubmission of an earlier submission. The following is a list of the peer review reports and author responses from that submission.

Round 1

Reviewer 1 Report

This manuscript submitted by Ramirez, Cepeda, Monroy, and Maldonado investigated the reaction between L-cysteine and 6-maleimidehxanoic acid and found the presence of tautomerism in the reaction. Overall, I found this article does not fit the topic of this special issue: How and Why to Investigate Multicomponent Reactions Mechanisms since it does not provide much mechanistic insight into the reaction. Moreover, I think this article can not be published in Molecule in this form due to the following reasons:

1) I found the introduction section in this paper very hard to read and follow. I suggest the authors add some figures/schemes/molecular structures into the background sections, especially where they describe the mechanisms. The authors should consider re-write the introduction. 

2) The authors should reference figure 1 in their manuscript.

3)I suggest the authors add at least one reaction equation to the results and discussions section.

4) the authors should add the NMR solvent to figure 4.

5) the authors should consider adding a 2D-NMR spectrum to the manuscript.

6) If the authors want to publish this work in molecule, I suggest them to consider including more substrates in order to investigate the effect of steric or functional groups to the reactivity/selectivity.

Reviewer 2 Report

This paper misses the mark, I'm sorry to say. The title promises tautomerism between S-linked and N-linked cysteine-maleimide adducts. "Tautomerism" might not be the best word -- perhaps isomerization or equilibration? -- but this is a moot point, because I think the structures are mis-assigned.

  1. The mechanism is implausible: the "cyclic intermediate" depicted in Figure 5 would have five bonds to carbon.
  2. If dynamic tautomerization (isomerization) were viable, then one would expect the separated isomers to re-equilibrate to the mixture upon standing. 

I think the two isomers are diastereomers arising from non-diastereoselective addition of cysteine. The addition of FMOC-cysteine could be diastereoselective, but more likely is that the isomers of this larger adduct are not as easily separated by chromatography.

An experiment that could shed light on this question is to add an additional equivalent of maleimide to the adducts. If the adducts are diastereomeric and N-linked, then they should react at similar rates to each other, and more slowly than the initial thiol-maleimide ligation. If the adducts are N-linked and S-linked isomers, then the N-linked isomer (bearing a free thiol) should react faster than the S-linked isomer.

Unless or until these central points can be cleared up, I do not support publication of this work.